# LEARNING INTERPRETABLE CHARACTERISTIC KERNELS VIA DECISION FORESTS

## ABSTRACT

Decision forests are widely used for classification and regression tasks. A lesser known property of tree-based methods is that one can construct a proximity matrix from the tree(s), and these proximity matrices are induced kernels. While there has been extensive research on the applications and properties of kernels, there is relatively little research on kernels induced by decision forests. We construct Kernel Mean Embedding Random Forests (KMERF), which induce kernels from random trees and/or forests using leaf-node proximity. We introduce the notion of an asymptotically characteristic kernel, and prove that KMERF kernels are asymptotically characteristic for both discrete and continuous data. Because KMERF is data-adaptive, we suspected it would outperform kernels selected *a priori* on finite sample data. We illustrate that KMERF nearly dominates current state-of-the-art kernel-based tests across a diverse range of high-dimensional two-sample and independence testing settings. Furthermore, our forest-based approach is interpretable, and provides feature importance metrics that readily distinguish important dimensions, unlike other high-dimensional non-parametric testing procedures. Hence, this work demonstrates the decision forest-based kernel can be more powerful and more interpretable than existing methods, flying in the face of conventional wisdom of the trade-off between the two.

## 1 INTRODUCTION

Decision forests, an ensemble method popularized by Breiman (2001), have proven highly effective in tasks involving classification and regression, particularly in high-dimensional scenarios (Caruana and Niculescu-Mizil, 2006; Caruana et al., 2008; Tomita et al., 2020a).

One noteworthy feature of random forests is their ability to generate a proximity matrix. Breiman (2002) defined one such matrix as the percentage of decision trees in which two observations end up in the same leaf node. This proximity matrix serves as an induced kernel or similarity matrix, quantifying the similarity between pairs of observations. Any random partition method can generate such a similarity matrix, and the link between random forests and induced kernels is established in the literature (Davies and Ghahramani, 2014; Scornet, 2015; Biau and Scornet, 2016).

Kernels find applications in various statistical and machine learning tasks (Schölkopf and Smola, 2002; Evgeniou et al., 2005; Hofmann et al., 2008; Cho and Saul, 2019). One fundamental statistical task is independence testing, where both kernel and distance-based statistics have been proposed, including Hilbert-Schmidt Independence Criterion (Hsic) (Gretton et al., 2005; 2012a) and distance correlation (Dcorr) (Székely et al., 2007; Székely and Rizzo, 2013). Interestingly, these two methods can be seen as equivalent because distances and kernels are interchangeable with proper transformations (Sejdinovic et al., 2013; Shen and Vogelstein, 2021).

When employing a characteristic kernel like the Gaussian kernel, or equivalently, a strong negative-type metric such as Euclidean distance (Lyons, 2013; 2018), the resulting dependence measure is universally consistent for testing independence against any distribution with finite second moments. This property makes Hsic and Dcorr superior to traditional linear correlations like Pearson's correlation and its rank-based variants (Pearson, 1895; Spearman, 1904; Kendall, 1970).

While Dcorr and Hsic are consistent and capable of detecting relationships when the sample size is sufficiently large, they exhibit low power in cases where the sample size is inadequate for the

complexity of the dependency. This can occur when dealing with strongly nonlinear dependencies, excessive noise in relationships, or high-dimensional data, which is particularly challenging for independence testing (Ramdas et al., 2015; Huang and Huo, 2017; Zhu et al., 2020).

The existing independence testing literature has shown that better finite sample testing power can be achieved through more data-adaptive approaches, such as utilizing local distances (Vogelstein et al., 2019; Shen et al., 2020) or adaptive kernel selection (Gretton et al., 2012b). As random forests produce a kernel and are renowned for their excellent performance with nonlinear and multivariate data, this paper investigates a decision forest-based kernel choice to enhance both the power and interpretability of the kernel statistical tests.

We demonstrate that, subject to specific conditions for partitions, the partition kernel attains the characteristic property, enabling it to uniquely represent every distribution. Notably, these conditions require the partitions to be sample-adaptive, introducing an additional layer of flexibility that enables better modeling of complex data structures.

Then we leverage a sample kernel based on decision forests and assess its performance in statistical hypothesis testing. This approach, called KMERF test, involves the fusion of several components, including random forest (Breiman, 2001), characteristic kernels (Gretton et al., 2005), unbiased test statistic transformations (Székely and Rizzo, 2014), and a fast chi-square approximation for a p-value (Shen et al., 2022). The resulting KMERF test achieves significant empirical advantages over existing methods for independence and k-sample testing, particularly in scenarios involving nonlinear and multivariate data.

Additionally, the method offers the capability to estimate feature importance through random forest, enhancing interpretability and providing rankings for important dimensions for multivariate data. To demonstrate the practical applicability of this approach, we apply it to real-world data to identify peptides closely associated with cancer. For additional background, simulation function details, complete theorem proofs, and additional simulation figures, are located in the appendix.

## 2 TECHNICAL BACKGROUND

**Definition 1.** *Given* $x_1, x_2 \in \mathbb{R}^p$, *and a set of partitions* $\Phi = \{\phi_m, m = 1, \ldots, M\}$ *of the space* $\mathbb{R}^p$. *We define the partition kernel* $K^\Phi : \mathbb{R}^p \times \mathbb{R}^p \to [0, 1]$ *as:*

$$K^\Phi(x_1, x_2) = \frac{1}{M} \sum_{m=1}^{M} [\mathbb{1}(\phi_m(x_1) = \phi_m(x_2))],$$

where $\mathbb{1}(\cdot)$ is the indicator function that checks whether the two observations lie in the same leaf region of each partition. Here, $M$ represents the number of partitions in the set $\Phi$, which is a fixed and positive number. For each partition, the resulting regions are referred to as the leaf region.

To illustrate this concept, consider a straightforward example: Suppose we have $M = 1$, a single partition denoted as $\phi_1$, and we are working with one-dimensional data ($p = 1$). Suppose $\phi_1$ divides the real number line $\mathbb{R}$ into two leaf regions via the origin, namely $(-\infty, 0)$ and $[0, +\infty)$. Consequently, $K^\Phi(x_1, x_2) = 1$ when $x_1$ and $x_2$ lie on the same side of the origin, and $K^\Phi(x_1, x_2) = 0$ when $x_1$ and $x_2$ lie on the opposite sides of the origin.

In a more general setting, the partition set consists of multiple partitions, and the kernel $K^\Phi(x_1, x_2) = 0$ if, for every partition from 1 to $M$, $x_1$ and $x_2$ do not belong to the same leaf region. Conversely, $K^\Phi(x_1, x_2) > 0$ if there exists at least one partition $\phi_m$ where $x_1$ and $x_2$ reside in the same leaf region of $\phi_m$.

The partition kernel always possesses the following property:

**Theorem 1.** *The partition kernel* $K^\Phi$ *is always positive definite, i.e., for any* $n \geq 2$, $x_1, \ldots, x_n \in \mathbb{R}^p$ *and* $a_1, \ldots, a_n \in \mathbb{R}$, *it satisfies*

$$\sum_{i,j=1}^{n} a_i a_j K^\Phi(x_i, x_j) \geq 0.$$

This theorem holds by the nature of the partition kernel, which can be expressed as the average of permuted block diagonal matrices. Each matrix originating from an individual partition is positive semidefinite (Davies and Ghahramani, 2014). Consequently, when adding together positive semidefinite matrices, the resulting matrix remains positive semidefinite.

## 3 PARTITION KERNEL

Next, we prove that, subject to specific conditions pertaining to the partitions, the partition kernel can indeed be considered characteristic. In other words, it has the ability to uniquely quantify the underlying distribution.

**Theorem 2.** *Suppose the partition set $\Phi$ satisfies the following: For any point $z \in \mathbb{R}^p$ and some positive value $\epsilon$ that defines a ball region $b(z, \epsilon)$, it holds that:*

1. *For any $x \notin b(z, \epsilon)$, $\phi_m(z) \neq \phi_m(x)$ for every $m = 1, \ldots, M$.*

2. *There exists a dense subset $\omega_z \subseteq b(z, \epsilon)$ such that for every $x \in \omega_z$, there exists at least one $m$ for which $\phi_m(z) = \phi_m(x)$.*

*Then the partition kernel $K^\Phi$ is a characteristic kernel. In other words, for any two continuous random variables $X_1$ and $X_2$ with probability distributions $F_{X_1}$ and $F_{X_2}$, the following equivalence holds:*

$$E[K^\Phi(\cdot, X_1)] = E[K^\Phi(\cdot, X_2)] \text{ if and only if } F_{X_1} = F_{X_2}.$$

In essence, a characteristic kernel implies that the mapping from the random variable $X$ to the expectation $E[K^\Phi(\cdot, X)]$ is injective, meaning that different distributions of $X$ result in distinct expectation values. While the theorem is stated for the continuous density, the discrete case is also presented in the proof, which requires a slight modification of the condition.

The conditions placed on the partition set require it to span the Euclidean space (or the support of the underlying distribution), while each leaf region remains dense. A straightforward example is a partition that evenly divides the interval $[0, 1]$ into $n$ sub-intervals. Consider $z = 0.01$ and $\epsilon = 0.01$; this partition satisfies the stated conditions for all values of $n$ greater than or equal to $50$.

These conditions shares similarities with the consistency property of k-nearest-neighbors (Devroye et al., 1996). It is well-known that the k-nearest-neighbor classifier is asymptotically Bayes optimal as both the number of samples ($n$) and the number of neighbors ($k$) tend to infinity, while the ratio $\frac{k}{n}$ approaches zero. If we view k-nearest-neighbor as a partition, where $\phi_1(x_1) = \phi_1(x_2)$ if and only if $x_2$ falls within the k-nearest-neighborhood of $x_1$, then as $n$ and $k$ tend to infinity and $\frac{k}{n}$ approaches zero, the conditions in Theorem 2 are satisfied.

Therefore, the k-nearest-neighbor induced partition kernel is "asymptotically" characteristic. It is asymptotic in the sense that the k-nearest-neighbor method is sample-based, and as such, the induced partition kernel is not characteristic for any finite $n$. Instead, it serves as an estimator that converges to the underlying population kernel that is inherently characteristic.

It is important to note that even for classical kernels like the Gaussian kernel, the concept of a characteristic kernel is fundamentally an asymptotic notion when applied to sample data. This is because the definition of a characteristic kernel relies on population expectations, which, in practical applications with sample data, are estimated using sample expectations of sample kernel matrices.

In contrast, the characteristic nature of the partition kernel inherently mandates the partitions to be sample-adaptive, much like the nearest-neighbor partition. This requirement necessitates the estimation of not only population expectations but also the population kernel itself. While this additional estimation step may introduce some variance into the estimation process, it endows the kernel with better modeling capabilities. This is achieved through data-adaptive partition construction, enabling the kernel to better capture complicates data structures. Such adaptability can be indispensable in a wide range of statistical and machine learning applications.

## 4 DECISION FOREST-INDUCED KERNEL FOR STATISTICAL TESTING

Random forest stands out as one of the most widely recognized partition methods, generating a set of partitions, with each partition referred to as a tree. Consequently, we direct our attention to the random forest-induced kernel, and leverage this induced kernel for testing independence. Our proposed approach, KMERF, involves the following key steps:

Given $\mathbf{X} = \{x_i \in \mathbb{R}^p, i = 1, \ldots, n\}$ and $\mathbf{Y} = \{y_i \in \mathbb{R}, i = 1, \ldots, n\}$,

1. **Train random forest**: Run random forest with $M$ trees on the dataset $(\mathbf{X}, \mathbf{Y})$, considering $\mathbf{X}$ as the predictor variable. Consequently, each tree generates a partition $\phi_m(\mathbf{X})$ for $\mathbb{R}^p$, and the entire forest forms the collection of partitions $\Phi(\mathbf{X})$.

2. **Estimate kernel**: Calculate the random forest-induced kernel for $\mathbf{X}$:

$$\tilde{K}_{ij}^{\Phi(\mathbf{X})} = \frac{1}{m} \sum_{m=1}^{M} [\mathbb{1}(\phi_m(x_i) = \phi_m(x_j))].$$

3. **Distance-kernel transformation**: Apply an unbiased kernel transformation (Szekely and Rizzo, 2014) on $\tilde{K}^{\Phi(\mathbf{X})}$. Namely, let

$$\mathbf{L}_{ij}^{\mathbf{X}} = \begin{cases} \tilde{K}_{ij}^{\Phi(\mathbf{X})} - \frac{1}{n-2} \sum_{t=1}^{n} \tilde{K}_{it}^{\Phi(\mathbf{X})} - \frac{1}{n-2} \sum_{s=1}^{n} \tilde{K}_{sj}^{\Phi(\mathbf{X})} + \frac{1}{(n-1)(n-2)} \sum_{s,t=1}^{n} \tilde{K}_{st}^{\Phi(\mathbf{X})} & i \neq j \\ 0 & i = j \end{cases}$$

4. **Calculate test statistic**: Let $K^{\mathbf{Y}}$ be any characteristic kernel for $\mathbf{Y}$, e.g., the Gaussian kernel or the Euclidean distance. Compute $\mathbf{L}_{ij}^{\mathbf{Y}}$ using the same unbiased transformation, followed by the test statistic

$$c_n(\mathbf{X}, \mathbf{Y}) = \frac{1}{n(n-3)} \text{trace}(\mathbf{L}^{\mathbf{X}} \mathbf{L}^{\mathbf{Y}}).$$

5. **Compute p-value**: Compute the p-value via the following chi-square approximation (Shen et al., 2022):

$$p = 1 - F_{\chi_1^2 - 1} \left( n \cdot \frac{c_n(\mathbf{X}, \mathbf{Y})}{\sqrt{c_n(\mathbf{X}, \mathbf{X}) \cdot c_n(\mathbf{Y}, \mathbf{Y})}} \right),$$

where $\chi_1^2$ is the chi-square distribution of degree 1. Reject the independence hypothesis if the p-value is less than a specified type 1 error level, say 0.05.

The KMERF method comprises three primary components: random forest and kernel computation, distance correlation computation, and hypothesis testing. The first step involves the application of the standard random forest from (Breiman, 2001), with a setting of $M = 500$. Additional details regarding the standard random forest are provided in the appendix. In the second step, the random forest-induced partition kernel is derived, which is equivalent to the proximity matrix obtained from the random forest. The third step involves the normalization of the random forest-induced kernel, followed by the fourth step, which computes the unbiased statistic similar to unbiased distance or kernel correlation (Szekely and Rizzo, 2014). Finally, in the fifth step, a chi-square test is applied, which has been demonstrated to approximate the null distribution of the unbiased statistic (Shen et al., 2022).

Note that an alternative process to compute a p-value for KMERF is through the permutation test, which is a standard procedure for testing independence (Good, 2005). Specifically, randomly permute the index of $\{y_i\}$, compute the new kernel for $\{x_i\}$ based on permuted $\{y_i\}$, then compute the permuted statistic. This process necessitates training a new random forest for each permutation and typically requires more than 100 permutations. As a result, it can be quite slow, especially for large sample sizes. On the other hand, the chi-square-based test has been shown to approximate the results of the permutation test effectively (Shen et al., 2022), making it the preferred choice for testing in this paper.

Considering the number of trees $M$, the number of dimensions $p$, and the number of samples $n$, the complexity of the random forest is $\mathcal{O}(Mpn \log n)$. The correlation computation complexity is

$\mathcal{O}(n^2)$, and the chi-square testing complexity is $\mathcal{O}(1)$. As a result, the overall process is fast and scalable. Finally, note that we assumed $y_i \in \mathbb{R}$ for simplicity. However, this method can be applied to arbitrary dimensions as well. In such cases, one may run a random forest between the entire $\mathbf{X}$ and each dimension of $\mathbf{Y}$, which simply adds additional partitions to the partition set.

## 5 VALID AND CONSISTENT TESTING

Given the underlying kernel is characteristic, the KMERF test is valid and consistent for testing independence, similar to DCorr and HSIC. We first formalize the independence hypothesis as follows: suppose $n$ samples of $(x_i, y_i) \overset{iid}{\sim} F_{XY}$, i.e., $x_i$ and $y_i$ are realizations of random variables $X$ and $Y$. The hypothesis for testing independence is

$$H_0 : F_{XY} = F_X F_Y,$$
$$H_A : F_{XY} \neq F_X F_Y.$$

A tightly related hypothesis is the k-sample task: let $u_i^j \in \mathbb{R}^p$ be the realization of random variable $U_j$ for $j = 1, \ldots, l$ and $i = 1, \ldots, n_j$. Suppose the $l$ datasets that are sampled i.i.d. from $F_1, \ldots, F_l$ and independently from one another. Then,

$$H_0 : F_1 = F_2 = \cdots = F_l,$$
$$H_A : \exists \, j \neq j' \text{ s.t. } F_j \neq F_{j'}.$$

It has been shown that by combining the $l$ datasets into a single dataset $\mathbf{X}$ and introducing an auxiliary random variable $\mathbf{Y}$, the k-sample hypothesis can be tested by any dependence measure (Panda et al., 2021).

**Theorem 3.** *Assuming each tree trained when computing the KMERF statistic $c_n$ satisfies the conditions in Theorem 2, then*

$$\lim_{n \to \infty} c_n(\mathbf{X}, \mathbf{Y}) = c \geq 0,$$

*with equality to 0 if and only if $F_{XY} = F_X F_Y$. Moreover, for sufficiently large $n$ and sufficiently small type 1 error level $\alpha$, the KMERF test is valid and consistent for independence or k-sample testing.*

Random forests exhibit a property similar to k-nearest-neighbors, and they do satisfy the conditions outlined in Theorem 2 in the following manner: as the number of observations $n$ increases towards infinity, and $k$ represents the number of observations in each leaf region, if $k$ increases and $\frac{k}{n} \to 0$, then each tree satisfies the conditions in Theorem 2.

In standard random forest, the number of observations in each leaf region is usually a fixed number, and the sample size is never truly infinite in practical applications. Consequently, the random forest-induced kernel $\tilde{K}^{\Phi(\mathbf{X})}$, in practice, does not precisely equal the underlying characteristic kernel $K^{\Phi(\mathbf{X})}$ but provides a sample estimate of it.

## 6 SIMULATIONS

In this section we exhibit the consistency and validity of KMERF and conduct a comprehensive simulation study to compare its testing power with other competing methods. We utilize the `hyppo` package in Python (Panda et al., 2020), which uses `scikit-learn` (Pedregosa et al., 2011) random forest with 500 trees and default hyper-parameters otherwise. The proximity matrix is calculated from this random forest, and subsequently, the KMERF statistic and p-value are computed following the process outlined in Section 4.

### 6.1 TESTING INDEPENDENCE

In this section we compare KMERF to Multiscale Graph Correlation (MGC) (Vogelstein et al., 2019), Distance Correlation (Dcorr) (Székely et al., 2007; Szekely and Rizzo, 2009), Hilbert-Schmidt Independence Criterion (Hsic) (Gretton et al., 2007), and Heller-Heller-Gorfine (HHG)

method (Heller et al., 2013), Canonical Correlation Analysis (CCA) (Hotelling, 1936), and the RV coefficient (Robert and Escoufier, 1976). The HHG method has been shown to work very well against nonlinear dependencies (Heller et al., 2013). The MGC method has been shown to work well against linear, nonlinear, and multivariate dependencies (Shen et al., 2020). The CCA and RV coefficients are multivariate extensions of Pearson correlation. For each method, we use the corresponding implementation in `hyppo` with default settings.

We take 20 multivariate simulation settings as specified in Vogelstein et al. (2019) (see Appendix C.1 for full details), consisting of various linear, monotone, and strongly nonlinear dependencies with $p$ increasing, $q = 1$, and $n = 100$. To estimate the testing power in each setting, we generate dependent $(x_i, y_i)$ for $i = 1, \ldots, n$. That is, each simulation function $f$, We then compute the test statistic for each method, and calculate how often the p-value is less than the type 1 error level of $\alpha = 0.05$. This is repeated for $r = 10000$ times. The results indicate that KMERF is frequently the top-performing method or one of the top performers in most scenarios, except for cases like the circle and ellipse simulations, as illustrated in Figure 1.

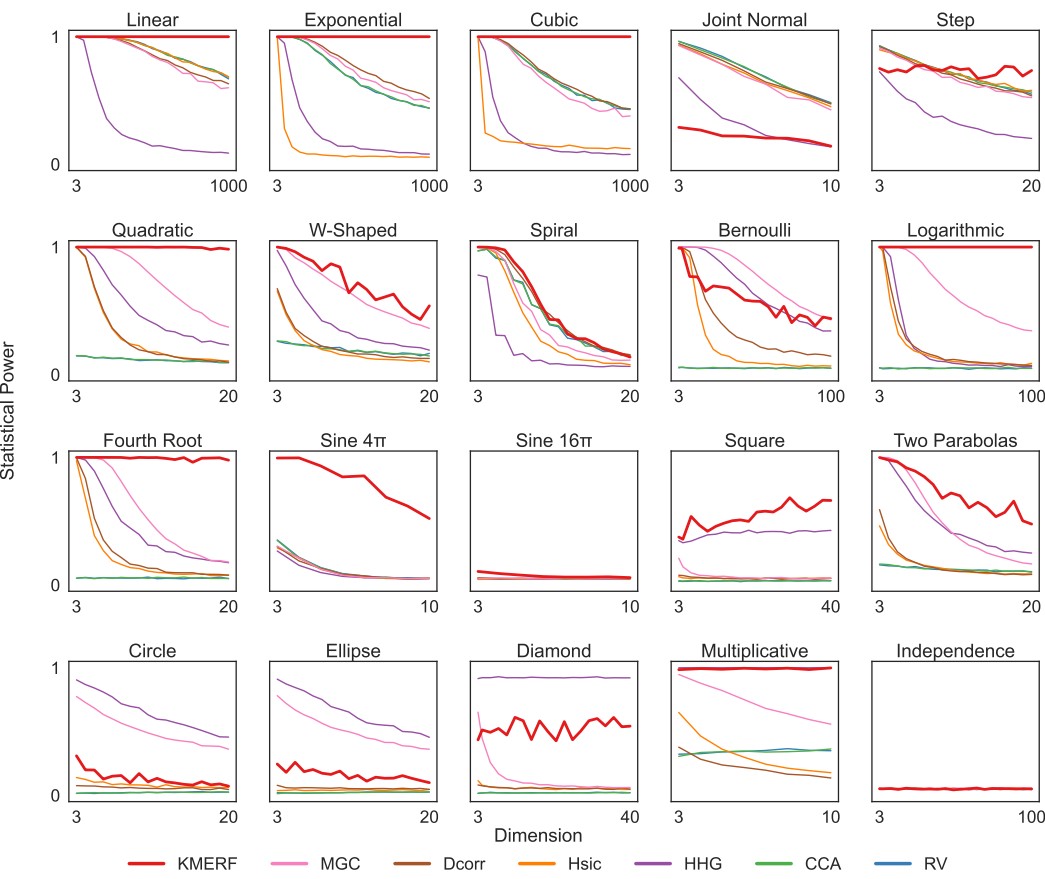

Figure 1: Multivariate independence testing power for 20 different settings (see Appendix C.1 for full details) with increasing $p$, fixed $q = 1$, and $n = 100$. For the majority of the simulations and simulation dimensions, KMERF performs as well as, or better than, existing multivariate independence tests in multivariate dependence testing.

## 6.2 Two Sample Testing

In this section, we compared the performance of different methods in a two-sample testing scenario using various simulation settings. The simulations included functions with different geometry, such as linear and nonlinear, monotonic and non-monotonic (see Appendix C.2 for full details). The

dimensionality varied from $p = 3$ to $p = 10$, with $q = 1$ and a sample size of $n = 100$. To create an independent second sample for testing, a random rotation was applied to the generated samples.

In Figure 2, we can observe that KMERF consistently performs as well as or better than other tests across the majority of simulation settings and dimensions. In complex data structures such as exponential, cubic, and trigonometric functions, the estimated kernel adapts more effectively to the underlying structure than existing kernels. As a result, KMERF demonstrates a substantial advantage in testing power in these scenarios.

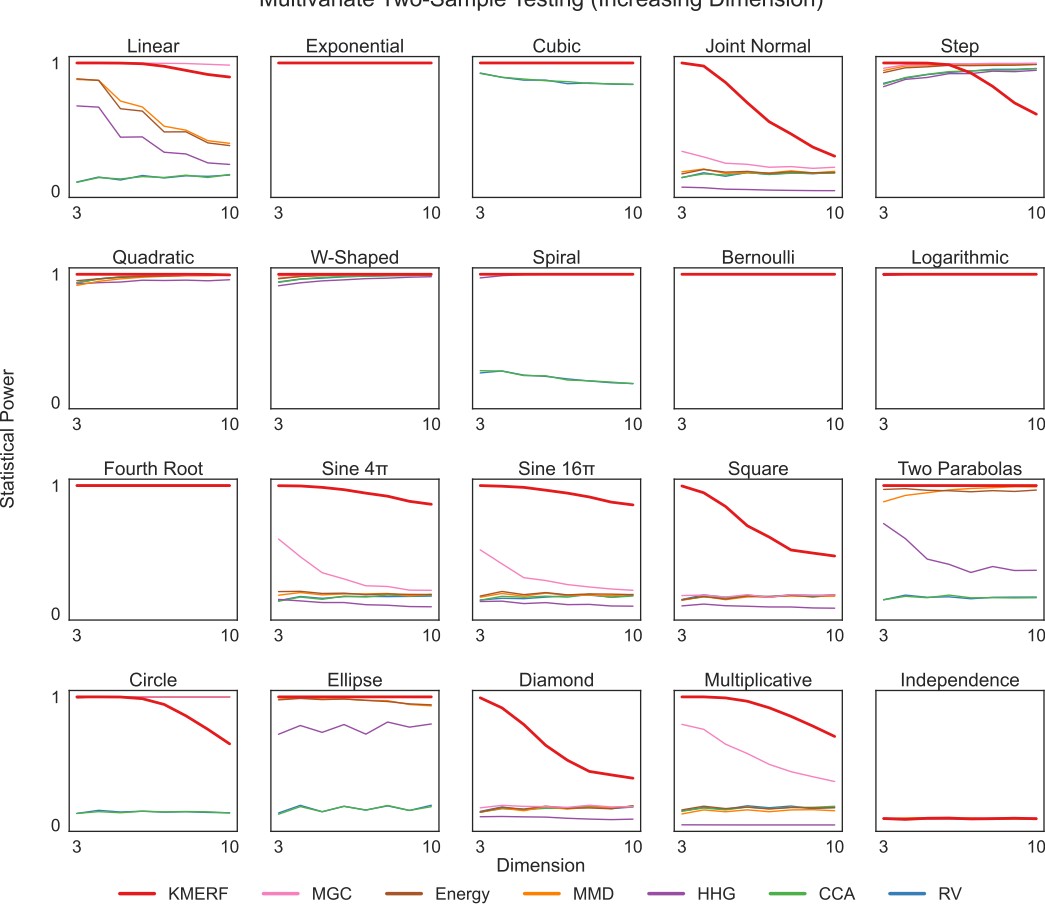

Figure 2: Multivariate two-sample testing power for 20 different settings (see Appendix C.2 for full details) with increasing $p$, fixed $q = 1$, and $n = 100$. For nearly all simulations and simulation dimensions, KMERF performs as well as, or better than, existing multivariate two-sample tests in multivariate dependence testing.

## 6.3 INTERPRETABILITY

KMERF not only demonstrates superior statistical power but also provides valuable insights into the importance of each feature within the dataset by estimating the Gini importance (Breiman, 2001). Figure 3 presents normalized 95% confidence intervals of Gini importance for each simulation. The black line represents the mean, and the light gray line shows the 95% confidence interval. Feature importance values were normalized using min-max feature scaling. In all simulations, the weights of the dimensions were designed to decrease, with $w_d = 1/d$, indicating that the first dimension has the highest importance while the remaining dimensions have diminishing importance. Remarkably, the figure demonstrates that the feature importance estimated by random forest closely approximates

the ground truth in most scenarios, with dimension 1 being identified as the most important feature and diminishing feature importance for the subsequent dimensions.

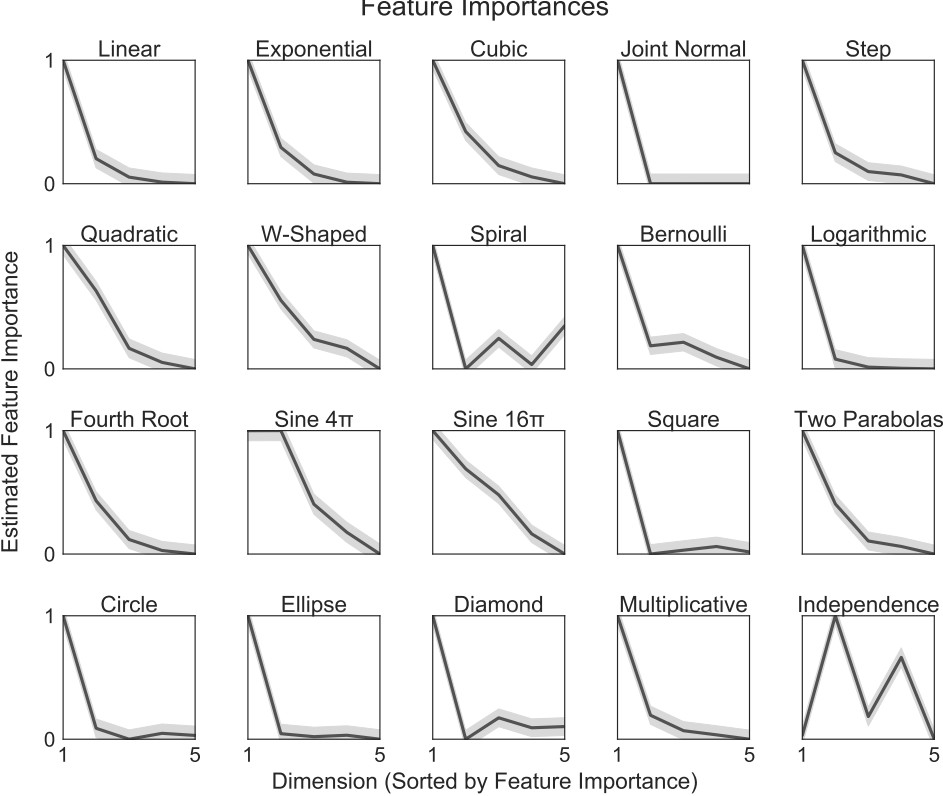

Figure 3: The figure displays the normalized mean (in black) and the corresponding $95\%$ confidence intervals (in light grey) achieved through min-max normalization. These values represent the relative feature importances computed using random forest across five dimensions in each simulation, each tested with 100 samples. As expected, the estimated feature importance diminishes with increasing dimensionality.

## 7 REAL DATA

We then applied KMERF to a dataset consisting of proteolytic peptides derived from blood samples of 95 individuals with pancreatic ($n = 10$), ovarian ($n = 24$), colorectal cancer ($n = 28$), and healthy controls ($n = 33$) (Vogelstein et al., 2019). The processed data included 318 peptides from 121 proteins (for full details, see Appendix F). The left side of figure 4 displays the p-values obtained by KMERF for the comparison between pancreatic and healthy subjects, as well as the p-values for the comparison between pancreatic cancer and all other subjects. The test identifies neurogranin as a potentially valuable marker for pancreatic cancer, which is supported by existing literature (Yang et al., 2015; Willemse et al., 2018). While some other tests also identified this biomarker, they also identified other markers upregulated in other types of cancers (Yang et al., 2015). The significant peptides for each test were then extracted and k-nearest-neighbor classification with leave-one-out cross validation was performed. As shown in the right side of figure 4, the peptide determined by KMERF achieves the best true and false positive rates for detecting pancreatic cancer.

## 8 DISCUSSION

Existing research has employed random forests for various hypothesis testing purposes, including F-tests, feature screening (Coleman et al., 2022), and two-sample testing (Hediger et al., 2022).

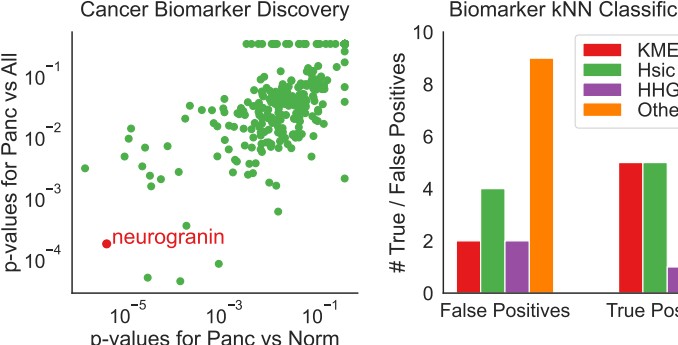

Figure 4: (Left) The KMERF p-value for each peptide, where KMERF successfully identifies a distinct protein. (Right) The counts of true and false positives through a leave-one-out $k$-nearest neighbor classification, with the best $k$ selected from the range $[1, 10]$. This classification only uses the significant peptides detected by each method. The peptide pinpointed by KMERF achieves the highest true and false positive rates. It is worth noting that both MGC and KMERF yield identical results by selecting the same unique peptide. On the other hand, Hsic and HHG identify two and three peptides, respectively, and we report the false positives and true positives specifically for these peptides. The remaining methods, namely Dcorr, CCA, and RV, do not identify any significant peptides. Consequently, the orange column represents the baseline performance when using all dimensions, resulting in 0 true positives and 9 false positives for this prediction task.

However, these studies have predominantly utilized other outputs of random forests instead of the proximity matrix. To the best of our knowledge, no prior research has delved into the characteristic properties of the partition kernel, nor has there been any investigation into the direct application of the random forest-induced kernel for statistical hypothesis testing.

The main contribution of this paper revolves around demonstrating the characteristic property of the partition kernel. The specified conditions emphasize that for the partition kernel to be characteristic, the partitions must be sample-adaptive, revealing novel insights that were previously unknown. Capitalizing on these insights has given rise to the KMERF test for assessing independence and conducting two-sample tests, which demonstrates distinct numerical advantages.

This manuscript offers a wealth of insights with significant implications. It notably expands the realm of potential kernel choices, showcasing the potency of sample-adaptive kernel selection through decision forests and paving the way for future explorations. Other kernels, whether derived from variations of the standard random forest (Davies and Ghahramani, 2014; Scornet, 2015) or from the proximity matrix of alternative random partition or forest algorithms (Geurts et al., 2006; Tomita et al., 2020b), can be directly integrated into this framework.

A previous method, the multiscale graph correlation (Vogelstein et al., 2019; Shen et al., 2020), can be viewed as a partition kernel that adapts to the data using the Euclidean distance. In multiscale graph correlation, distant distances are discarded, and only nearby distances are retained, essentially creating a partition. As random forest can be seen as a nearest neighbor algorithm (Lin and Jeon, 2006), the forest-induced kernel can be considered a natural extension of Vogelstein et al. (2019). This extension offers more data-adaptive estimates of nearest neighbors by leveraging supervised information.

Furthermore, establishing that the random forest-induced kernel is characteristic represents a crucial step toward constructing lifelong learning kernel machines with robust theoretical foundations (Pentina and Ben-David, 2015; Vogelstein et al., 2023). It would be valuable to delve deeper into the underlying theoretical mechanisms of the induced characteristic kernel and to assess the performance of these forest-induced kernels in other machine learning problems, including classification, regression, clustering, and embedding.

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

# A  ADDITIONAL BACKGROUND

## A.1  RANDOM FOREST AND THE PROXIMITY KERNEL

Given a dataset of $(x_i, y_i)$ for $i = 1, \ldots, n$, we employed the standard Classification and Regression Trees (CART) algorithm (Breiman, 1984; Hastie et al., 2001) to construct each tree. In the context of an independence test where $y_i$ is continuous, the construction of the tree involves identifying the optimal dimension $j$ and the corresponding optimal split point $s$ that minimizes the following objective function:

$$\min_{j,s} \left[ \min_{c_a} \sum_{x \in R_a(j,s)} (y_i - c_a)^2 + \min_{c_b} \sum_{x \in R_b(j,s)} (y_i - c_b)^2 \right].$$

Here, $c_a$ and $c_b$ are the sample means within the regions $R_a$ and $R_b$ respectively. These regions can be thought of as half-planes within the current region. For example, for the initial split, $R_a = \{x_i | x_i^j \le s\}$ and $R_b = \{x_i | x_i^j > s\}$.

In essence, the CART algorithm identifies the optimal pair $(j, s)$ to iteratively build a tree structure on the sample data, aiming to decrease an objective function within each region until a predefined stopping criterion is met. By default, the stopping criterion is five observations within each region for regression trees. In classification, the algorithm typically stops when each region is 'pure,' meaning it contains samples from a single class. Upon reaching the stopping criterion, the resulting regions within the tree are commonly referred to as leaf nodes.

# B  PROOFS

**Theorem 1.** *The partition kernel $K^\Phi$ is always positive definite, i.e., for any $n \ge 2$, $x_1, \ldots, x_n \in \mathbb{R}^p$ and $a_1, \ldots, a_n \in \mathbb{R}$, it satisfies*

$$\sum_{i,j=1}^n a_i a_j K^\Phi(x_i, x_j) \ge 0.$$

*Proof.* We prove this theorem by considering the sample kernel matrix. Consider $x_1, x_2, \ldots, x_n \in \mathbb{R}^p$, and a set of partitions $\Phi = \{\phi_m, m = 1, \ldots, M\}$ of the space $\mathbb{R}^p$. The partition kernel, based on the definition in the main paper, satisfies:

$$K^\Phi(x_i, x_j) = \frac{1}{M} \sum_{m=1}^M [\mathbb{1}(\phi_m(x_1) = \phi_m(x_2))],$$

for any $i, j$ from 1 to $n$.

We can view $K^\Phi$ as the $n \times n$ sample kernel matrix, which is the average of each individual partition kernel $K^{\phi_m}$. Moreover, each individual kernel matrix can be represented as a permuted block-diagonal matrix, with each block being a matrix of ones, and zeros elsewhere Davies and Ghahramani (2014). It follows that

$$K^\Phi = \frac{1}{M} \sum_{m=1}^M K^{\phi_m}$$

$$= \frac{1}{M} \sum_{m=1}^M Q_m B_m Q_m^T.$$

Here $B_m$ is a block diagonal matrix with each block representing a leaf region, and $Q_m$ is a permutation matrix that reorders sample indices to group them in the same leaf region. For example, when each leaf region only contains one observation, $B_m$ becomes the identity matrix; when $x_1$ and $x_3$ are in the same leaf region, the first $2 * 2$ submatrix of $B_m$ is a matrix of ones, while $Q_m$ permutes the second index with the third index.

Since each block matrix $B_m$ is always positive semidefinite, and permutation does not change eigenvalues, each $B_m$ remains positive semidefinite after permutation. As the summation of positive semidefinite matrices is still positive semidefinite, $\mathbf{K}^{\mathbf{X}}$ is always positive semidefinite.

Consequently, we have

$$\sum_{i,j=1}^{n} a_i a_j K^{\Phi}(x_i, x_j) \geq 0,$$

so a partition kernel is always positive definite. $\qquad\square$

**Theorem 2.** *Suppose the partition set $\Phi$ satisfies the following: For any point $z \in \mathbb{R}^p$ and some positive value $\epsilon$ that defines a ball region $b(z, \epsilon)$, it holds that:*

1. *For any $x \notin b(z, \epsilon)$, $\phi_m(z) \neq \phi_m(x)$ for every $m = 1, \ldots, M$.*

2. *There exists a dense subset $\omega_z \subseteq b(z, \epsilon)$ such that for every $x \in \omega_z$, there exists at least one $m$ for which $\phi_m(z) = \phi_m(x)$.*

*Then the partition kernel $K^{\Phi}$ is a characteristic kernel. In other words, for any two continuous random variables $X_1$ and $X_2$ with probability distributions $F_{X_1}$ and $F_{X_2}$, the following equivalence holds:*

$$E[K^{\Phi}(\cdot, X_1)] = E[K^{\Phi}(\cdot, X_2)] \text{ if and only if } F_{X_1} = F_{X_2}.$$

*Proof.* The if direction is straightforward. When $F_{X_1} = F_{X_2}$, it is evident that for any $z \in \mathbb{R}^p$, we always have

$$E[K^{\Phi}(z, X_1)] = E[K^{\Phi}(z, X_2)], \tag{1}$$

such that

$$E[K^{\Phi}(\cdot, X_1)] = E[K^{\Phi}(\cdot, X_2)]. \tag{2}$$

To prove the only if direction, it suffices to find a point $z$ such that when $F_{X_1} \neq F_{X_2}$, we have

$$E[K^{\Phi}(z, X_1)] \neq E[K^{\Phi}(z, X_2)]. \tag{3}$$

Without loss of generality, let us consider a point $z$ and a ball-region centered at $z$ with radius $\epsilon > 0$ denoted by $b(z, \epsilon)$, where the density satisfies $f_{X_1}(x) > f_{X_2}(x)$ for all $x \in b(z, \epsilon)$. Such a region must exist when $f_{X_1}(x)$ is a continuous density.

Based on the conditions on the partition set, $K^{\Phi}(z, x) = 0$ for any $x \notin b(z, \epsilon)$, and there must exist a dense subset $\omega_z \subseteq b(z, \epsilon)$ such that $K^{\Phi}(z, x) > 0$ for all $x \in \omega_z$. As a result,

$$E[K^{\Phi}(z, X_1) - K^{\Phi}(z, X_2)]$$
$$= \int_x K^{\Phi}(z, x)(f_{X_1}(x) - f_{X_2}(x))dx$$
$$= \int_{x \notin b(z, \epsilon)} K^{\Phi}(z, x)(f_{X_1}(x) - f_{X_2}(x))dx + \int_{x \in b(z, \epsilon)} K^{\Phi}(z, x)(f_{X_1}(x) - f_{X_2}(x))dx$$
$$> 0,$$

where the first integral in line three is zero, and the second integral in line three is positive. Therefore, there always exists $z$ such that when $F_{X_1} \neq F_{X_2}$, $E[K^{\Phi}(z, X_1)] \neq E[K^{\Phi}(z, X_2)]$, and the partition kernel is characteristic.

The previous proof addresses the continuous case. When the density $f_{X_1}(x)$ is discrete or a mix of continuous and discrete, and there is no $b(z, \epsilon)$ such that $f_{X_1}(x) > f_{X_2}(x)$ for all $x \in b(z, \epsilon)$, then $z$ must exist on a discrete part, *i.e.*, there exists $z$ such that $Prob(X_1 = z) > Prob(X_2 = z) \geq 0$ (It's worth noting that $Prob(X_2 = z)$ may still be a density).

In this case, there may no longer exist a dense subset $\omega_z \subseteq b(z, \epsilon)$ such that $K^{\Phi}(z, x) > 0$. However, we no longer need the dense subset. All we need is the partitions to handle discrete case different and enforce $K^{\Phi}(z, x) = 0$ for any $x \neq z$. Note that $K^{\Phi}(z, z) = 1$ trivially, and

$$E[K^{\Phi}(z, X_1) - K^{\Phi}(z, X_2)] = (Prob(X_1 = z) - Prob(X_2 = z)) > 0.$$

Therefore, the discrete case also hold by an additional condition on the partitions: for any $z \in \mathbb{R}^p$ that is discrete on either density, we require that $\phi_m(z, x) = 0$ for any $x \neq z$ and every $m$. □

**Theorem 3.** *Assuming each tree trained when computing the KMERF statistic $c_n$ satisfies the conditions in Theorem 2, then*

$$\lim_{n \to \infty} c_n(\mathbf{X}, \mathbf{Y}) = c \geq 0,$$

*with equality to 0 if and only if $F_{XY} = F_X F_Y$. Moreover, for sufficiently large $n$ and sufficiently small type 1 error level $\alpha$, the KMERF test is valid and consistent for independence or k-sample testing.*

*Proof.* When $\tilde{K}^{\Phi(\mathbf{X})}$ satisfies the conditions, it is characteristic as proven in Theorem 2. The kernel choice for $\mathbf{Y}$ is also characteristic. Therefore, by Gretton et al. (2005); Lyons (2013), $c^n(\mathbf{x}, \mathbf{y})$ is asymptotically 0 if and only if independence.

By Shen et al. (2022), for sufficiently large $n$, the chi-square distribution $\frac{\chi_1^2 - 1}{n}$ dominates the true null distribution of the unbiased correlation in upper tail. Therefore, when $X$ and $Y$ are independent, the testing power is no more than the type 1 error level $\alpha$, making it a valid test. Conversely, when $X$ and $Y$ are dependent, the distribution $\frac{\chi_1^2 - 1}{n}$ converges to 0 in probability, while $c_n(\mathbf{X}, \mathbf{Y}) > 0$. As a result, the p-value tends to 0 and the testing power converges to 1, establishing it as a consistent test. □

## C  SIMULATIONS

### C.1  INDEPENDENCE SIMULATIONS

For the independence simulation, we test independence between $X$ and $Y$. For the random variable $X \in \mathbb{R}^p$, we denote $X_{|d|}, d = 1, \ldots, p$ as the $d^{th}$ dimension of $X$. $w \in \mathbb{R}^p$ is a decaying vector with $w_{|d|} = 1/d$ for each $d$, such that $w^\mathsf{T} X$ is a weighted summation of all dimensions of $X$. Furthermore, $\mathcal{U}(a, b)$ denotes the uniform distribution on the interval $(a, b)$, $\mathcal{B}(p)$ denotes the Bernoulli distribution with probability $p$, $\mathcal{N}(\mu, \Sigma)$ denotes the normal distribution with mean $\mu$ and covariance $\Sigma$, $U$ and $V$ represent some auxiliary random variables, $\kappa$ is a scalar constant to control the noise level, and $\epsilon$ is sampled from an independent standard normal distribution unless mentioned otherwise.

1. Linear$(X, Y) \in \mathbb{R}^p \times \mathbb{R}$:
$$X \sim \mathcal{U}(-1, 1)^p,$$
$$Y = w^\mathsf{T} X + \kappa \epsilon.$$

2. Exponential$(X, Y) \in \mathbb{R}^p \times \mathbb{R}$:
$$X \sim \mathcal{U}(0, 3)^p,$$
$$Y = \exp(w^\mathsf{T} X) + 10\kappa \epsilon.$$

3. Cubic$(X, Y) \in \mathbb{R}^p \times \mathbb{R}$:
$$X \sim \mathcal{U}(-1, 1)^p,$$
$$Y = 128\left(w^\mathsf{T} X - \frac{1}{3}\right)^3 + 48\left(w^\mathsf{T} X - \frac{1}{3}\right)^2 - 12\left(w^\mathsf{T} X - \frac{1}{3}\right) + 80\kappa \epsilon.$$

4. Joint Normal$(X, Y) \in \mathbb{R}^p \times \mathbb{R}^1$: Let $\rho = 1/2p$, $I_p$ be the identity matrix of size $p \times p$, $J_p$ be the matrix of ones of size $p \times 1$, and $\Sigma = \begin{bmatrix} I_p & \rho J_p \\ \rho J_p & (1 + 0.5\kappa) I_1 \end{bmatrix}$. Then,
$$(X, Y) \sim \mathcal{N}(0, \Sigma).$$

5. Step Function$(X, Y) \in \mathbb{R}^p \times \mathbb{R}$:

$$X \sim \mathcal{U}(-1, 1)^p,$$

$$Y = \mathbb{I}(w^\mathsf{T} X > 0) + \epsilon,$$

where $\mathbb{I}$ is the indicator function; that is, $\mathbb{I}(z)$ is unity whenever $z$ is true, and $0$ otherwise.

6. Quadratic$(X, Y) \in \mathbb{R}^p \times \mathbb{R}$:

$$X \sim \mathcal{U}(-1, 1)^p,$$

$$Y = (w^\mathsf{T} X)^2 + 0.5\kappa\epsilon.$$

7. W-Shape$(X, Y) \in \mathbb{R}^p \times \mathbb{R}$: For $U \sim \mathcal{U}(-1, 1)^p$,

$$X \sim \mathcal{U}(-1, 1)^p,$$

$$Y = 4\left[\left((w^\mathsf{T} X)^2 - \frac{1}{2}\right)^2 + \frac{w^\mathsf{T} U}{500}\right] + 0.5\kappa\epsilon.$$

8. Spiral$(X, Y) \in \mathbb{R}^p \times \mathbb{R}$: For $U \sim \mathcal{U}(0, 5)$, $\epsilon \sim \mathcal{N}(0, 1)$,

$$X_{|d|} = U \sin(\pi U) \cos^d(\pi U) \text{ for } d = 1, ..., p - 1,$$

$$X_{|p|} = U \cos^p(\pi U),$$

$$Y = U \sin(\pi U) + 0.4p\epsilon.$$

9. Uncorrelated Bernoulli$(X, Y) \in \mathbb{R}^p \times \mathbb{R}$: For $U \sim \mathcal{B}(0.5)$, $\epsilon_1 \sim \mathcal{N}(0, I_p)$, $\epsilon_2 \sim \mathcal{N}(0, 1)$,

$$X \sim \mathcal{B}(0.5)^p + 0.5\epsilon_1,$$

$$Y = (2U - 1) w^\mathsf{T} X + 0.5\epsilon_2.$$

10. Logarithmic$(X, Y) \in \mathbb{R}^p \times \mathbb{R}^1$: For $\epsilon \sim \mathcal{N}(0, I_p)$,

$$X \sim \mathcal{N}(0, I_p),$$

$$Y_{|d|} = 2 \log_2(|X_{|d|}|) + 3\kappa\epsilon_{|d|} \text{ for } d = 1.$$

11. Fourth Root$(X, Y) \in \mathbb{R}^p \times \mathbb{R}$:

$$X \sim \mathcal{U}(-1, 1)^p,$$

$$Y = |w^\mathsf{T} X|^{1/4} + \frac{\kappa}{4}\epsilon.$$

12. Sine Period $4\pi(X, Y) \in \mathbb{R}^p \times \mathbb{R}^1$: For $U \sim \mathcal{U}(-1, 1)$, $V \sim \mathcal{N}(0, 1)^p$, $\theta = 4\pi$,

$$X_{|d|} = U + 0.02pV_{|d|} \text{ for } d = 1$$

$$Y = \sin(\theta X) + \kappa\epsilon.$$

13. Sine Period $16\pi(X, Y) \in \mathbb{R}^p \times \mathbb{R}^1$: Same as above except $\theta = 16\pi$ and the noise on $Y$ is changed to $0.5\kappa\epsilon$.

14. Square$(X, Y) \in \mathbb{R}^p \times \mathbb{R}^1$: For $U \sim \mathcal{U}(-1, 1)$, $V \sim \mathcal{U}(-1, 1)$, $\epsilon \sim \mathcal{N}(0, 1)^p$, $\theta = -\frac{\pi}{8}$,

$$X_{|d|} = U \cos(\theta) + V \sin(\theta) + 0.05p\epsilon_{|d|},$$

$$Y_{|d|} = -U \sin(\theta) + V \cos(\theta).$$

15. Diamond$(X, Y) \in \mathbb{R}^p \times \mathbb{R}^p$: Same as above except $\theta = \pi/4$.

16. Two Parabolas$(X, Y) \in \mathbb{R}^p \times \mathbb{R}$: For $\epsilon \sim \mathcal{U}(0, 1)$, $U \sim \mathcal{B}(0.5)$,

$$X \sim \mathcal{U}(-1, 1)^p,$$

$$Y = \left((w^\mathsf{T} X)^2 + 2\kappa\epsilon\right) \cdot \left(U - \frac{1}{2}\right).$$

17. `Circle`$(X, Y) \in \mathbb{R}^p \times \mathbb{R}$: For $U \sim \mathcal{U}(-1, 1)^p$, $\epsilon \sim \mathcal{N}(0, I_p)$, $r = 1$,

$$X_{|d|} = r \left( \sin \left( \pi U_{|d+1|} \right) \prod_{j=1}^{d} \cos \left( \pi U_{|j|} \right) \right) \text{ for } d = 1, ..., p-1,$$

$$X_{|p|} = r \left( \prod_{j=1}^{p} \cos \left( \pi U_{|j|} \right) \right),$$

$$Y = \sin \left( \pi U_{|1|} \right) + 0.4 \epsilon_{|p|}.$$

18. `Ellipse`$(X, Y) \in \mathbb{R}^p \times \mathbb{R}^1$: Same as above except $r = 5$.

19. `Multiplicative Noise`$(x, y) \in \mathbb{R}^p \times \mathbb{R}^1$: $u \sim \mathcal{N}(0, I_p)$,

$$x \sim \mathcal{N}(0, I_p),$$

$$y_{|d|} = u_{|d|} x_{|d|} \text{ for } d = 1.$$

20. `Multimodal Independence`$(X, Y) \in \mathbb{R}^p \times \mathbb{R}$: For $U \sim \mathcal{N}(0, I_p)$, $V \sim \mathcal{N}(0, I_p)$, $U' \sim \mathcal{B}(0.5)^p$, $V' \sim \mathcal{B}(0.5)^p$,

$$X = U/3 + 2U' - 1,$$

$$Y = V/3 + 2V' - 1.$$

Figure F1 visualizes these equations. The light grey points in the figure are each simulation with noise added and the dark grey points are each simulation without noise added. Note that the last two simulations don't have any noise parameters.

## C.2 TWO-SAMPLE SIMULATIONS

We perform two-sample testing between $Z$ and $Z'$, generated as follows: let $Z = [X|Y]$ be the respective random variables from the independence simulation setup. Then define $Q_\theta$ as a rotation matrix for a given angle $\theta$, *i.e.*,

$$Q_\theta = \begin{bmatrix} \cos\theta & 0 & \dots & -\sin\theta \\ 0 & 1 & \dots & 0 \\ \vdots & \vdots & \ddots & \vdots \\ \sin\theta & 0 & \dots & \cos\theta \end{bmatrix}$$

Then we let

$$Z' = Q_\theta Z^\mathsf{T}$$

be the rotated versions of $Z$. Therefore, $Z$ and $Z'$ are equivalent in distribution when the rotation is an identity matrix, or if $Z$ itself is rotation-invariant.

Figure F2 visualizes the above simulations. The simulations light grey points is a simulated data set and the dark grey points are the same dataset rotated by 10 degrees counter-clockwise. Simulations are plotted with min-max normalization for visualization purposes.

## D EFFECT OF POST-INFERENCE FEATURE SELECTION

As shown in Figure 1 and Figure 2, the test power equals the type 1 error under independence. This supports the empirical validity of the test, which means the test will not cause an inflation of the false positive rate during post-selection inference. To further substantiate this point, we conducted an additional simulation, the results of which are presented in Figure F3.

We generated $(x_1, x_2, x_3, x_4, x_5, x_6)$, where each is drawn independently from a $\mathcal{U}(-1, 1)$ distribution. Then we set $Y = x_1 + x_2^2 + x_3 + \epsilon$, where $\epsilon$ is a noise parameter. This establishes a relationship between the first three variables and $Y$, while the last three variables are independent of $Y$. This is repeated for 100 replicates, and in each replicate we generate sample data, then compute the KMERF statistic and p-value between each of $x_1, \ldots, x_6$ and $Y$. We set the true positive rate

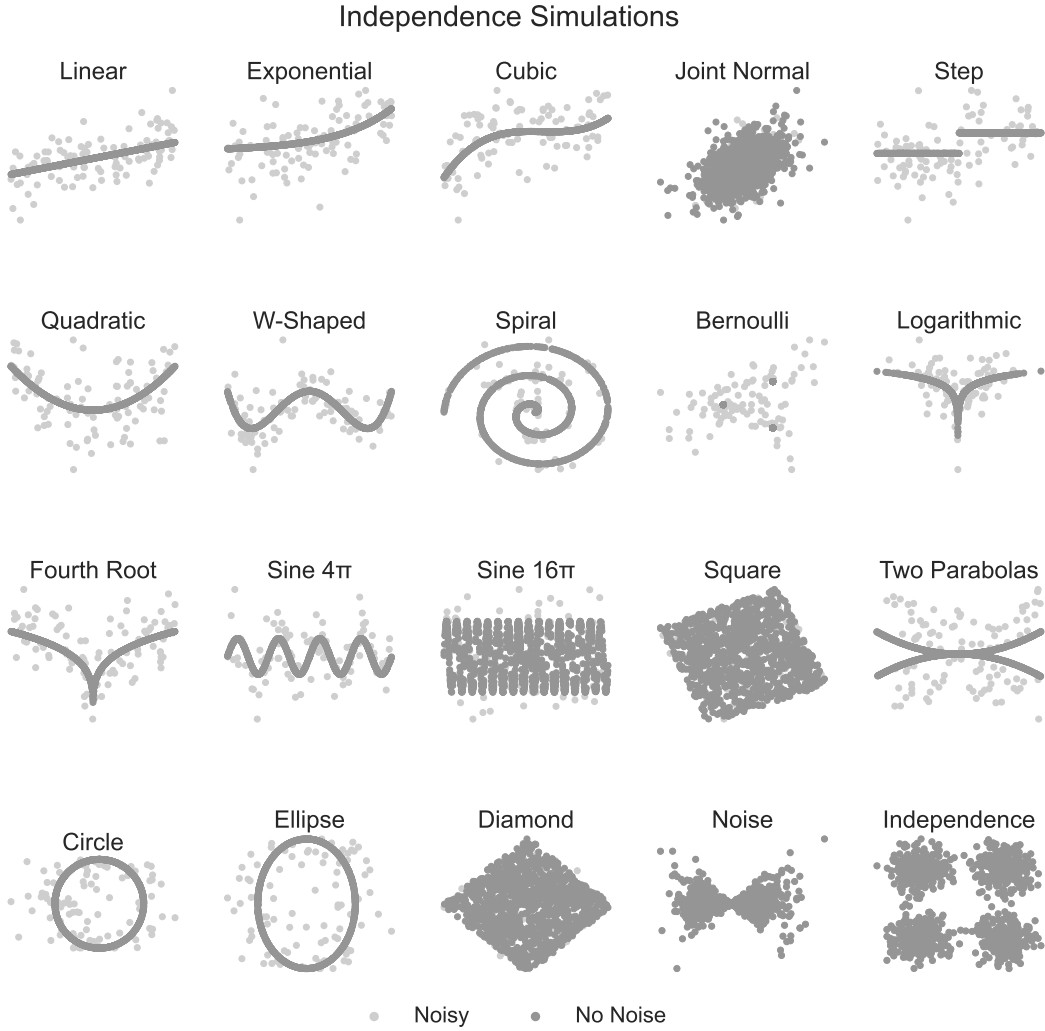

Figure F1: Simulations used for Figures 1 and 3. 100 points from noisy simulations (light grey points) on 1000 points from simulations without noise (dark grey points) for each of the 20 dimensional simulations shown above.

as how often the dependent variables are flagged as significant ($p < 0.05$), and set the false positive rate as how often the independent variables are flagged as significant. The computations were performed for each variable, and we report the average true positive among the first three variables, and the average false positive among the last three variables. As expected, the figure shows that the true positive goes to 1 and the false positive stays at 0.05.

# E  EFFECT OF NUMBER OF TREES

The number of trees has relatively small effect on the testing power of KMERF. To show this, we simulated the linear, spiral, and independence simulations from Appendix C.1 with $n = 100$ samples and $p = 10$. We then varied the number of trees $m$ from 10 to 100. As depicted in Figure F4, it is evident that $m$ has negligible impacts on the testing power of KMERF for these relationships and choices of $m$.

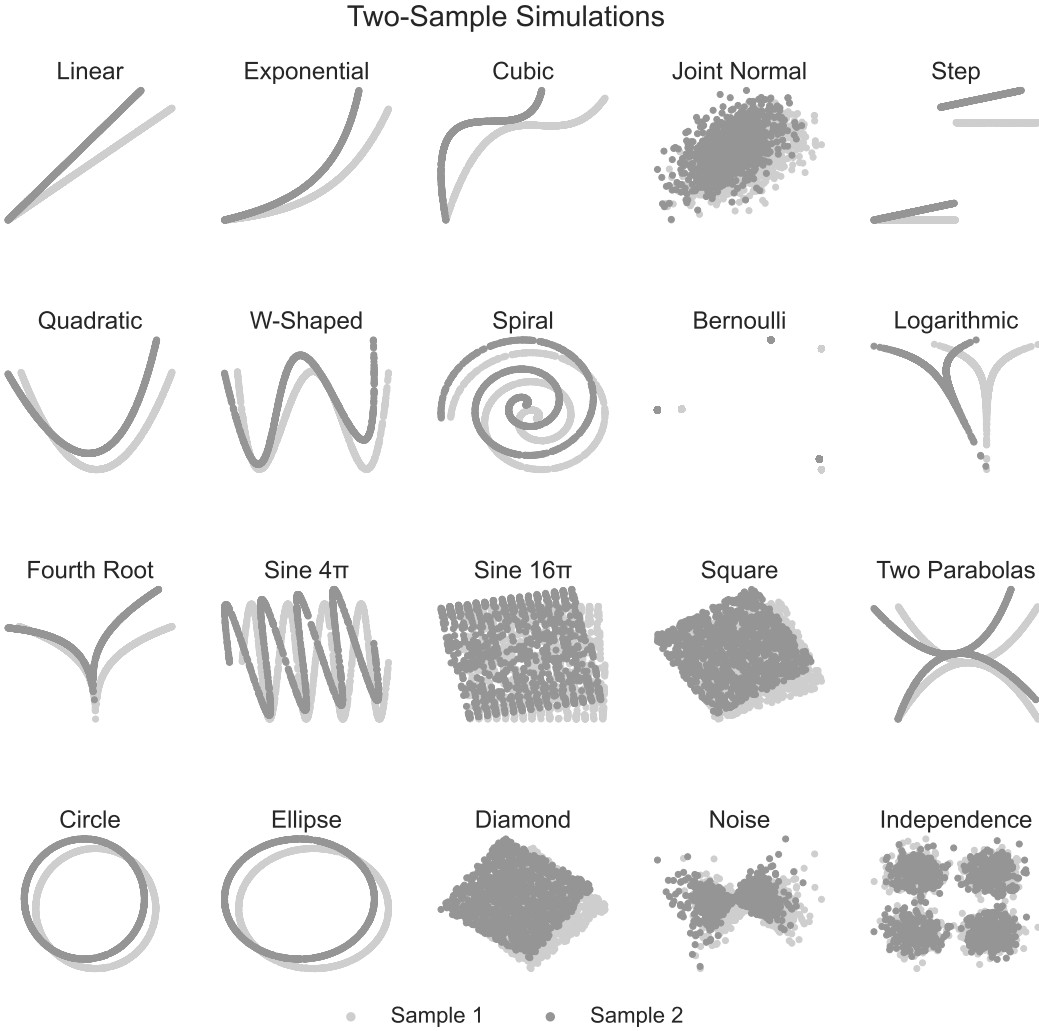

Figure F2: Simulations used for Figure 2. The first dataset (black dots) is 1000 samples from each of the 20 two-dimensional, no-noise simulation settings. The second dataset is the first dataset rotated by 10 degrees counter-clockwise. Simulations were normalized using min-max normalization for visualization purposes.

# F  REAL DATA

Previous studies have shown the utility of selection reaction monitoring when measuring protein and peptide abundance (Wang et al., 2011), and one was used to identify 318 peptides from 33 normal, 10 pancreatic cancer, 28 colorectal cancer, and 24 ovarian cancer samples (Wang et al., 2017). For our tests, we created a binary label vector, where 1 indicated the presence of pancreatic cancer in the patients, and 0 indicated its absence. We then evaluated the results at a type 1 error level of $\alpha = 0.05$ and applied the Benjamini-Hochberg procedure (Benjamini and Hochberg, 1995) to control the false discovery rate for our set of 318 p-values.

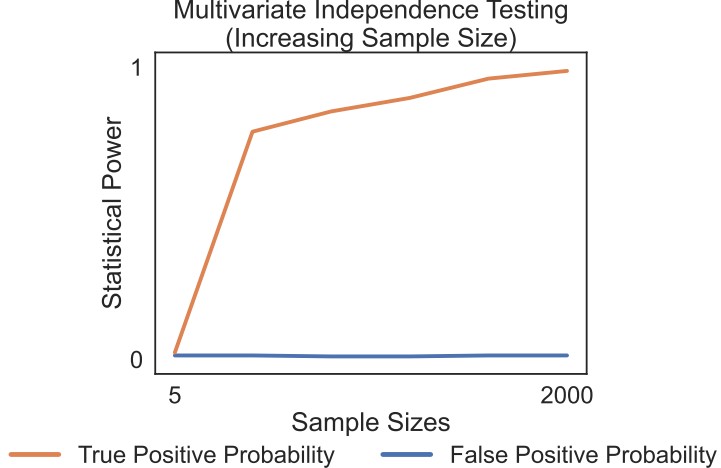

Figure F3: This figure illustrates an additional post-inference feature selection simulation performed using KMERF. The simulation setting is as follows: For variables $x_1, x_2, x_3, x_4, x_5, x_6 \sim \mathcal{U}(-1, 1)$ and random noise $\epsilon \in \mathcal{N}(0, 1)$, we let $Y = x_1 + x_2^2 + x_3 + 0.5\epsilon$. As a result, the first three variables are related to Y while the latter three do not. For each sample size and replicate, p-values $KMERF(x_1, y), \ldots, KMERF(x_6, y)$ are computed. This process is repeated for 100 replicates at each sample size. The average true positive rate converges to 1 and the false positive rate converges to 0.

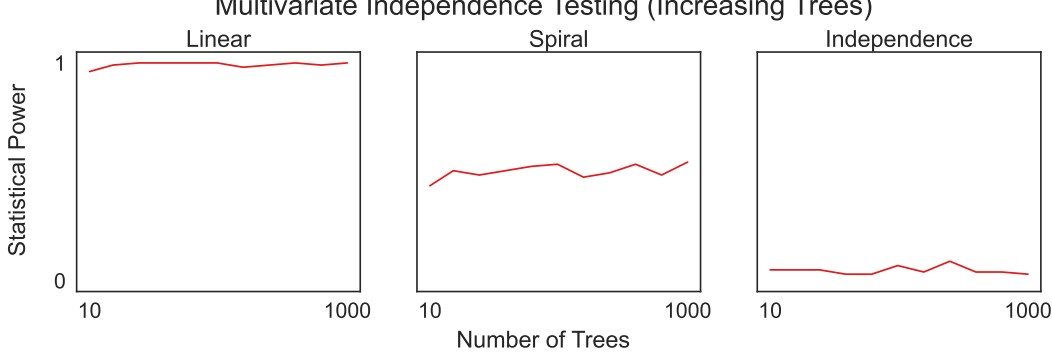

Figure F4: This figure illustrates how the testing power is influenced by the number of trees, considering $m$ within the range of 10 to 1000, and the parameters are set as $p = 10$ and $n = 100$ using the same linear, spiral, and independence relationships in the main paper. The number of trees has minimal impact on the testing power of KMERF.

