# OpenReview forum: "Learning Interpretable Characteristic Kernels via Decision Forests"
_ICLR.cc/2024/Conference — ICLR 2024 Conference Withdrawn Submission_

### Official Review · Reviewer_j4BZ · 2023-10-19

**Soundness:** 3 good
**Presentation:** 3 good
**Contribution:** 2 fair
**Rating:** 3
**Confidence:** 3

**Summary:**

This paper introduces Kernel Mean Embedding Random Forests (KMERF), a technique rooted in forest methodologies that formulates kernels via leaf-node proximity. The paper proves that KMERF kernels are asymptotically characteristic for discrete and continuous data. Furthermore, it is demonstrated that KMERF nearly surpasses the current state-of-the-art kernel-based tests across various scenarios. The approach is interpretable, offering feature importance metrics.

**Strengths:**

The paper is readable and well-written. The approach taken towards the objective is logical, and it is a solid contribution. I also appreciated that not only were numerical experiments conducted, but theoretical validity was also substantiated.

**Weaknesses:**

While the paper presents a solid contribution, I am yet to grasp its significance.
Although there is limited prior research on the characteristic properties of the partition kernel and the direct use of the Random Forest kernel for statistical hypothesis testing, just applying an existing kernel could be seen as a small contribution.
I think this paper's main strength potentially lies in the theoretical demonstration of the kernel's properties. However, the theory, based on straightforward proof, doesn't seem major advancement.
The mention of interpretability, attributed to forest properties, merely relies on using the existing kernel, and thus doesn't stand out as a unique contribution by the authors.
Consequently, despite its solid contributions, considering the paper to surpass the ICLR threshold seems difficult.

**Questions:**

1: During the experiments, the hyperparameters of Random Forest are fixed, but how would changes to these affect the results? I would like to know the extent of the impact that model selection has on experimental outcomes.

2: While Random Forest requires a target for training as it is a supervised learning model, I wonder if there are instances when performing independence testing of distributions where a target may not be present. What would be done in such cases?

3: Please let me know if there are any unique features or points regarding the KMERF algorithm that can be highlighted.

---

### Official Review · Reviewer_SxFq · 2023-10-27

**Soundness:** 3 good
**Presentation:** 3 good
**Contribution:** 2 fair
**Rating:** 5
**Confidence:** 4

**Summary:**

The paper introduces the kernel induced by the partition set pertinent to the trained random forest into the realm of the independence test. It shows under certain conditions the kernel yields valid and consistent tests. Experiments of independence testing, two-sample testing, and interpretability in terms of feature importance are conducted mainly on simulations.

**Strengths:**

1.	The paper has an original application of partition kernel to independence test and two-sample test, the result compared to the other methods are superior on simulation data;
2.	The kernel provides interpretability in terms of the feature importance.
3.	The partition kernel itself may find broader applications in the machine learning community; how to identify a characteristic kernel can be also very important to the learning theory.

**Weaknesses:**

1.	The contribution of the paper, mainly the introduction of the partition kernel into the independence test, is inadequate.
2.	Theorem 1 was proved in [lemma 3.1, Davies and Ghahramani, 2014].
3.	Simulation should also include classification problems (where Y is (with or without) class label).

**Questions:**

About the theory:
1.	The proof of theorem 2 uses the local dense condition (condition 2) to deduce the positivity of the second integral in the second equality, but the proof does not show why the kernel must be positive at the limit point of the dense set \omega_z.
2.	It is still not clear presented why random forest-induced partition kernel satisfies the conditions of Theorem 2, can you make the relevant claim “Random forests exhibit a property similar to k-nearest-neighbors, and they do satisfy the conditions outlined in Theorem 2” more rigorously?
3.	Previous methods such as HSIC and MGC choose kernels or some fixed rule (e.g. the knn rule which is not learned from data) a priori, independent of the data and thus independent of the independent test, while the proposed partition kernel learns rule from the particular sample, and the corresponding independence test then seems conditional on some learned information. Note that the independence tests are correlated to the performance of the random forest: observing that the KMERF has bad performance on the datasets such as the high-dimensional noised circle/diamond sample on which RF also has difficulties in classification or regression tasks. This is a little bit confusing to me. Does this learning procedure impair the validity of the independence test?

To make your content self-contained and more reader-friendly, please:
1.	Add more explanations on steps 3-5 in section 4, i.e., tell the reader why you do it;
2.	Include a mathematical introduction of the methods in comparison (MGC, HHG, HSIC, etc.);

Some problems with respect to the experiments:
1.	The noise scaling \kappa in your simulation is not specified;
2.	Figures from F1 and F2 are largely blank.

---

### Official Review · Reviewer_sjX1 · 2023-10-31

**Soundness:** 3 good
**Presentation:** 3 good
**Contribution:** 2 fair
**Rating:** 5
**Confidence:** 4

**Summary:**

This paper proposes a kernel method based on random forests with asymptotic properties and applies it to various high-dimensional two-sample and independence testing scenarios.

**Strengths:**

S1. The writing expression of this paper is clear, and the theory and experiment are easy to read.

S2. The research motivation of this paper is relatively clear. Due to the fact that adaptive kernels are generally superior to prior kernels in practical applications, and existing kernel methods generally lack interpretability, the random forest algorithm is used to construct partition kernels for testing problems.

**Weaknesses:**

W1. The research method of this paper lacks innovation. Kernel methods based on random forests are not uncommon, and this paper is more likely to directly apply them to hypothesis testing problems. The asymptotic properties of the complete version of the random forest are still an unsolved mystery, often with some simplified RF being proven to have asymptotic consistency.

W2. The experimental section shows that the proposed method is significantly superior to existing testing methods, but the author should explain the reasons for RF advantages through some visual experiments.

**Questions:**

Q1. I hope the author can analyze the differences in kernel characteristics between RF and other comparison methods under different data types.

---

### Official Review · Reviewer_m1gD · 2023-11-02

**Soundness:** 2 fair
**Presentation:** 3 good
**Contribution:** 3 good
**Rating:** 3
**Confidence:** 4

**Summary:**

The authors propose the use of random forests to generate a set of partitions that defines a kernel. They show that, under certain conditions, such partition-based kernels are characteristic and therefore valid and consistent for independence hypothesis testing. They demonstrate on synthetic data with known dependence that, in most cases, their independence hypothesis test has greater sensitivity compared to prior methods. It also provides an estimate of feature importance that aligns with the underlying simulation. They further demonstrate their methods' application to the identification of a relevant biomarker for pancreatic cancer.

**Strengths:**

The document is largely well organized and written. The proposed methods are practical, grounded, and motivated both theoretically and empirically. The simulated experiments are thorough, include many baselines, and clearly demonstrate the value of the proposed methods.

**Weaknesses:**

The problem addressed by the methods is not well-motivated in the introduction. Moreover, with no related works section, the work is not contextualized well with respect to prior work. A better discussion of prior work would help to understand the novel elements and the gap that they aim to fill. Also, I am not very familiar with the topic of independence testing, so it is difficult for me to judge novelty without more context. There are some ambiguities in the theoretical presentation, and I have doubts about the soundness of the theoretical claims, which the authors describe as the main contribution of the paper. With very limited evaluation on real-world data, it is hard to know how the successes on synthetic data will transfer to realistic applications.

**Questions:**

I am including here both questions and suggestions for improvement.

Theorem 1 is never referenced. Why is it important?

I have several questions and doubts about Theorem 2:
- The statement regarding $z$ and $\epsilon$ is somewhat ambiguous. Do you mean "For any point $z\in\mathbb{R}^p$, there exists a positive value $\epsilon$" or "For any point $z\in\mathbb{R}^p$ and positive value $\epsilon$"?
	- If the former, I think that the statement following (3) in the proof is not valid because the conditions only hold for one particular value of epsilon, which may not be an epsilon such that $f_{X_1}(x)>f_{X_2}(x)$ on all $x\in b(z,\epsilon)$.
	- If the latter, I cannot see how this condition can possibly be satisfied for all $\epsilon>0$, e.g. if if condition 1 holds for $\epsilon$, then condition 2 does not hold for $2\epsilon$.
- It would be nice to define a "dense subset", and perhaps make it clear why it is used (instead of a simpler condition). I had to look up this term, and I suspect other interested readers would too.
- The proof assumes that the probability densities are continuous. This must be stated in the theorem.
- It is also not clear that the given example (partition [0,1] into $n$ sub-intervals) is correct. For example, even for only the given $z=0.01$ and $\epsilon=0.01$, for a partition with any $n>50$, it is not true that $x$ and $z$ belong to the same leaf for all $x$ in some dense subset of $b(z,\epsilon)=[0,0.02]$.
- Likewise, I do not think it is possible, in general, to construct a partition such that, for any $x_1$ and $x_2$, $x_1$ and $x_2$ are in the same leaf if and only if $x_2$ is in the $k$-nearest-neighborhood of $x_1$.

It would be helpful to state the hypothesis being tested at the start of section 4 instead of section 5.

Distance-kernel transformation:
what is the purpose? It is hard to tell what it's doing by the equation alone.
How efficiently can we compute the distance-kernel transformation?

The chi-squared test is used as a more efficient alternative to permutation testing. What is the tradeoff? That is, what is the disadvantage of the chi-squared approach, and how severely can we expect it to affect the results?

What is the complexity of step 2, computing K?

I don't know how fair it is to claim that the algorithm is overall "fast and scalable". Mnp could be very large, and the n^2 term can become problematic for very large data. How does this compare to prior work in this are, e.g. the experimental baselines? Also, it would help if the experiments supported the claim of scalability, but the fixed n=100 does not inspire confidence.

In Theorem 3, it is stated that each tree must individually satisfy the conditions of Theorem 2. Then why is Theorem 2 stated as a property of sets of partitions, and not of individual partitions?

Second-to-last paragraph of section 5: I am hoping for some clarification on Theorem 2 as described above, but as to my current understanding, I don't see how the properties listed here imply that the conditions are satisfied. Moreover, it is certainly not true, without some assumptions about $F_{XY}$, that $k/n \to 0$ for random forests, unless you modify the tree growth algorithm to ensure that leaves have a sufficiently *small* sample count.

Why do the results show statistical power vs. dimension instead of vs. sample count? Shouldn't this be necessary to experimentally demonstrate consistency?

What value of $\kappa$ was used for the data simulations?

If you can, an explanation of why KMERF performs poorly on the few data sets where it does would be useful.

Only one real-world data set is considered, and the experimental setup is somewhat confusing. What exactly are the hypotheses being tested on the two axes of the left part of Figure 4?

With regards to the interpretability experiment:
it looks like you use fixed $p=5$ here, and the x axis in Figure 3 is the feature index? If so, the former should be stated explicitly, and the x axis label and caption of Figure 3 (which suggest that the x axis is p) should be corrected. Also, when you say "sorted by feature importance", I think you mean sorted by decreasing coefficient in the underlying simulation? Maybe there is a different way to say that, but it is not to be confused with the estimated feature importance.

Nitpicks:
- Definition 1: should be one sentence
- Definition 1: either define how a set $\phi_m$ is used as a function, or write it using common notation such as $\in$
- Definition 1: state that the members of a partition are called "leaves" before using the term "leaf"
- I don't think the example in the paragraph following Definition 1 is particularly necessary or helpful.
- For the stated $O(Mpn \log n)$ complexity for fitting a random forest, we have to assume the trees are approximately balanced. If they are highly unbalanced, it is $O(Mpn^2)$.